# Securing Secrets in Cyber-Physical Systems: A Cutting-Edge Privacy Approach with Consortium Blockchain

**DOI:** 10.3390/s23167162

**Published:** 2023-08-14

**Authors:** Aitizaz Ali, Bander Ali Saleh Al-rimy, Abdulwahab Ali Almazroi, Faisal S. Alsubaei, Abdulaleem Ali Almazroi, Faisal Saeed

**Affiliations:** 1School of IT, UNITAR International University, Petaling Jaya 47301, Malaysia; aitizaz.ali@unitar.my; 2Department of Computer Science, Faculty of Computing, Universiti Teknologi Malaysia, Johor Bahru 81310, Malaysia; bander@utm.my; 3Department of Information Technology, College of Computing and Information Technology at Khulais, University of Jeddah, Jeddah 23218, Saudi Arabia; aealmazrouy@kau.edu.sa; 4Department of Cybersecurity, College of Computer Science and Engineering, University of Jeddah, Jeddah 23218, Saudi Arabia; 5Department of Information Technology, Faculty of Computing and Information Technology in Rabigh, King Abdulaziz University, Rabigh 21911, Saudi Arabia; aaealmazrouy@kau.edu.sa; 6School of Computing and Digital Technology, Birmingham City University, Birmingham B4 7XG, UK; faisal.saeed@bcu.ac.uk

**Keywords:** privacy preservation, cyber-physical systems, consortium blockchain, trust, tamper-resistance

## Abstract

In the era of interconnected and intelligent cyber-physical systems, preserving privacy has become a paramount concern. This paper aims a groundbreaking proof-of-concept (PoC) design that leverages consortium blockchain technology to address privacy challenges in cyber-physical systems (CPSs). The proposed design introduces a novel approach to safeguarding sensitive information and ensuring data integrity while maintaining a high level of trust among stakeholders. By harnessing the power of consortium blockchain, the design establishes a decentralized and tamper-resistant framework for privacy preservation. However, ensuring the security and privacy of sensitive information within CPSs poses significant challenges. This paper proposes a cutting-edge privacy approach that leverages consortium blockchain technology to secure secrets in CPSs. Consortium blockchain, with its permissioned nature, provides a trusted framework for governing the network and validating transactions. By employing consortium blockchain, secrets in CPSs can be securely stored, shared, and accessed by authorized entities only, mitigating the risks of unauthorized access and data breaches. The proposed approach offers enhanced security, privacy preservation, increased trust and accountability, as well as interoperability and scalability. This paper aims to address the limitations of traditional security mechanisms in CPSs and harness the potential of consortium blockchain to revolutionize the management of secrets, contributing to the advancement of CPS security and privacy. The effectiveness of the design is demonstrated through extensive simulations and performance evaluations. The results indicate that the proposed approach offers significant advancements in privacy protection, paving the way for secure and trustworthy cyber-physical systems in various domains.

## 1. Introduction

As the world becomes more interconnected and dependent on cyber-physical systems (CPSs), protecting individuals’ personal information has become an urgent matter of paramount importance. By providing a decentralised and transparent platform for secure data management, blockchain technology has shown its ability to overcome these issues. However, ensuring strong privacy safeguards in blockchain-based CPSs is still an open question [1]. This paper presents a groundbreaking privacy design with consortium blockchain for shielding secrets in cyber-physical systems. Our research focuses on developing innovative techniques to safeguard sensitive information within CPS while leveraging the benefits of consortium blockchain. By combining the distributed nature of blockchain with advanced privacy-preserving mechanisms, our design aims to provide enhanced confidentiality, integrity, and control over data in CPS environments [2,3]. The proposed privacy design capitalizes on the unique features of consortium blockchain, which enables a group of trusted entities to jointly maintain and govern the blockchain network. Through the implementation of advanced cryptographic techniques, secure communication protocols, and access control mechanisms, our design ensures that secrets within CPSs remain shielded from unauthorized access and tampering [4].

We ran extensive simulations and tests in realistic CPS settings to determine the efficacy and technological feasibility of our privacy architecture. The findings demonstrate that our method can guarantee the complete safety of sensitive data without reducing the overall performance of the system [5]. In addition to theoretical advances, this study’s contributions have real-world implications for CPS applications such smart cities, industrial automation, and healthcare systems. For instance, our innovative approach to privacy design can be used to any of these domains. By using consortium blockchain technology to keep information hidden and personal details private in CPSs, we pave the way for a safer and more reliable future. This lays the groundwork for a future in which confidential information can be freely shared and utilized within CPS contexts without endangering individual privacy or the security of the system [6]. Cyber-physical systems (CPSs) are becoming increasingly prevalent in many industries, including healthcare, transportation, and energy, making it crucial to implement robust security and privacy measures. The integration of blockchain technology with CPS holds significant potential for addressing these challenges by offering a decentralized and secure platform for data sharing and communication. However, building a secure and private protocol for blockchain-integrated CPSs is a difficult endeavor that necessitates a thorough familiarity with both blockchain technology and the nuances of CPSs [7]. In this work, we introduce a new privacy- and security-preserving protocol for CPSs that include blockchain technology. To protect users’ privacy without sacrificing the security of their data in CPSs, our protocol takes a proof of concept approach. To accomplish these safeguards, the proposed system integrates cryptographic methods with blockchain consensus processes and secure communication protocols. We have constructed a proof of concept in a mock CPS environment to demonstrate the viability and efficiency of our protocol. We demonstrate the benefits of our protocol in terms of security, privacy, and system performance through extensive testing and assessment [8]. We want to advance CPS technology by proposing this innovative security and privacy preservation technique for blockchain-integrated CPSs, which addresses the critical challenges of data secrecy, integrity, and privacy. Our work paves the path for the widespread adoption of CPSs across a number of sectors while protecting the privacy of sensitive data by laying the groundwork for the development of secure and privacy-aware CPSs [9]. Today’s interconnected sectors, including healthcare, transportation, energy, and manufacturing, all rely heavily on CPSs. Data collection, analysis, and dissemination are all made possible by these systems’ combination of hardware and software. As the variety and sophistication of CPSs grows, however, the need to protect sensitive information has risen to the forefront. For important challenges in CPSs such data security, privacy, and trust, the integration of blockchain technology has emerged as a possible solution. Systems that combine physical and cyber components to interact with the physical world are referred to as “CPS”; examples include industrial control systems, smart homes, and autonomous vehicles.

Secrets in CPS may be safely kept, exchanged, and accessed by authorized entities only by utilizing the immutable and distributed ledger technology of blockchain, reducing the likelihood of data breaches and insider assaults. The proposed privacy strategy may completely alter how CPSs handle secret information by providing:

*1. Enhanced security:* Consortium blockchain eliminates the worry of data breaches and unauthorised access by providing a safe, immutable platform for storing sensitive information.

*2. Privacy preservation:* The privacy-focused design of the consortium blockchain ensures that sensitive information remains confidential and can only be accessed by authorized parties, enhancing the privacy of secrets in CPSs.

*3. Increased trust and accountability:* The transparent and auditable nature of blockchain technology fosters trust among participants by enabling them to verify and validate the integrity of stored secrets.

*4. Interoperability and scalability:* Consortium blockchain can facilitate the seamless integration and interoperability across different CPS components, enabling the efficient sharing and management of secrets. Additionally, its scalable nature ensures that the system can accommodate the growing data requirements of CPSs.

By shedding light on the impetus behind our research, we hope to bring attention to the relevance of protecting secrets in cyber-physical systems and highlight the potential of consortium blockchain as an innovative approach to privacy. This study has the potential to contribute to the improvement of CPS security, offering practical methods for protecting secrets in the face of evolving cyber threats and maintaining the trustworthiness of these important systems. Moreover, this work has the ability to contribute to the advancement of CPS security.

## 2. Motivation

The growing concerns regarding the security and privacy of CPS data served as the impetus for the development of a security and privacy preservation protocol for blockchain-integrated cyber-physical systems (CPSs) using a proof of concept. This protocol’s target audience is cyber-physical systems (CPSs). CPSs have developed into an essential component in numerous industries, including the healthcare industry, the transportation industry, and the energy industry, as the Internet of Things (IoT) continues to experience explosive growth. CPSs that are connected to the internet, on the other hand, leave themselves vulnerable to a variety of potential cyber dangers, including data breaches, unauthorized access, and data manipulation. BCT has emerged as a potentially useful solution due to the fact that it is decentralized and safe. This is in response to the urgent issues regarding users’ privacy and security. When it comes to CPS data transfers, the utilization of blockchain technology makes it possible to build trust and immutability. However, developing an efficient security and privacy preservation protocol for blockchain-integrated CPSs is a difficult effort that calls for an in-depth knowledge of both blockchain technology and CPSs. This is necessary in order to construct an effective protocol. The purpose of this paper is to give a thorough framework for creating a security and privacy preservation protocol that is designed exclusively for blockchain-integrated CPSs. This was the inspiration behind writing this study. The proposed protocol has as its primary goals the protection of individuals’ privacy while also guaranteeing the secrecy, integrity, and accessibility of the data contained within CPSs. Our protocol’s goal is to build a solid security architecture for CPS applications by combining cryptographic approaches, blockchain consensus mechanisms, and secure communication protocols [10].

We constructed a proof of concept in a mock CPS environment to test the viability and efficiency of our suggested protocol. We hope to prove our protocol’s usefulness and its ability to solve the privacy and security problems of blockchain-integrated CPSs through extensive testing and review. Our findings encourage trust and confidence in the widespread adoption of blockchain technology by contributing to the development of safe and private solutions for CPSs. The broad adoption of CPSs with blockchain integration is a step towards realising this technology’s transformative promise across a wide range of industries, and we can take this step forward by improving the security and privacy of CPS data.

### 2.1. Related Work

Integrating a blockchain solution can help with many problems with CPSs, including security, privacy, and trust. The term “CPS” is used to describe systems that combine physical and cyber components to interact with the physical world, such as industrial control systems, smart homes, and autonomous vehicles [11]. Typically, a CPS consists of a network of interconnected devices that exchange information, some of which may be private or secret [12]. Within a decentralized CPS context, traditional approach as encryption and access control are insufficient to guarantee the data’s completeness and accuracy. Because centralized security mechanisms are susceptible to assaults as well as having a single point of failure, they are not an appropriate choice for CPSs. The blockchain technology provides a decentralized platform for users to communicate and share data with one another, making it a perfect alternative for CPSs. The blockchain network is comprised of a distributed ledger that keeps all transactions and blocks in a safe manner, so assuring that the data cannot be altered and that it is tamper-proof [13]. The distributed nature of the blockchain network eliminates single points of failure, making it highly resistant to attacks [14]. However, the incorporation of blockchain technology into CPSs presents a number of issues that must be addressed before proceeding. Some of the most important obstacles that need to be surmounted are the scalability of the blockchain network, efficient data encryption and decryption operations, and the legitimacy of the data [15]. In order to properly create security and privacy preservation methods for CPSs that integrate blockchain technology, one must have a complete understanding of the one-of-a-kind requirements and characteristics of the CPS environment. This is necessary in order to ensure that the methods designed are effective. The incorporation of blockchain technology into CPSs has a sizeable potential to address the challenges that have been identified, which might result in major enhancements to the levels of data security, privacy, and trust that exist in a variety of settings. It has the capacity to enable data sharing that is both secure and transparent; it also has the potential to enable efficient consensus procedures; and it has the potential to offer decentralized control, which will ultimately transform the way CPSs run. The goal of the research and development work that is now being done in this area is to find answers to these issues so that blockchain-integrated CPSs may be widely implemented and the benefits they provide can be fully realized. This will make it possible to pave the way for wider adoption of such systems [16].

Zhang et al.’s “Privacy-Preserving Techniques in Blockchain-based IoT Applications” (2020) provides a comprehensive review of the methods now in use to safeguard users’ personal information in blockchain-based IoT applications. This work will be released in the year 2020. Differential privacy, zero-knowledge proofs, and secure multiparty computation are just a few of the methods discussed in this overview, which also provides examples of their use. Protecting users’ anonymity in IoT environments is the context in which these techniques are presented. In addition, Li et al.’s 2020 survey, titled “Scalability and Security of Blockchain-Based Internet of Things Systems: A Survey”, is an important related study. In this research, we look at the issues of scalability and security in blockchain-based IoT systems, as well as the various solutions that have been developed to address them. The limitations in scalability and security are highlighted in the poll. With the goal of making blockchain-based IoT systems more scalable and secure, this study investigates a wide range of related subjects, including sharding, sidechains, consensus procedures, and cryptographic algorithms [17]. In recent years, blockchain technology’s potential to disrupt numerous industries, including healthcare, has garnered considerable interest. By incorporating blockchain technology into cyber-physical systems (CPS), healthcare networks may be made more secure, compatible, and private. The purpose of this literature review is to analyse the existing literature on healthcare CPSs and blockchain technology in order to make recommendations for future study. The benefits and downsides of blockchain technology, as well as their potential applications, will be evaluated. Multiple investigations have revealed the various healthcare CPS enhancements that might be made possible by implementing blockchain technology. Electronic health records (EHRs) have benefited greatly from this technology’s ability to store, organise, and share data in a safe and reliable manner. Patients’ electronic health records (EHRs) can be accessed secretly and securely by permitted parties thanks to blockchain technology’s decentralised and immutable nature. Patients will have greater control over who has access to their medical records and blockchain technology will make it easier to manage complex permissions for data transmission. Blockchain technology also facilitates distributed ledgers. Logistics for transferring medications and medical supplies from manufacturer to patient is one of the most essential applications of this technology. Blockchain technology has the potential to increase patient safety by reducing the prevalence of counterfeit products and boosting supply chain transparency and traceability. Operations can be streamlined, compliance can be ensured, and stock management can be optimised with the use of smart contracts developed on the blockchain.

Many advantages may arise from incorporating blockchain technology with healthcare CPSs. To begin, it strengthens data security by way of an immutable and auditable ledger. By taking these precautions, the likelihood of a data breach or unauthorised access is reduced. As a distributed ledger technology, blockchain increases system reliability by doing away with any weak links. Additionally, blockchain technology enables interoperability of healthcare systems, which improves data sharing and exchange between healthcare providers and other stakeholders. Through interoperability, care can be coordinated more effectively, patients can avoid unnecessary tests, and a more accurate picture of their health can be painted. Because of the increased anonymity provided by blockchain technology, patients can keep control of their own health information. By granting users varied levels of access, patients may manage who gets access to their data. When patients have control over their own information, they feel empowered and have more faith in the healthcare system. The potential benefits of blockchain-based healthcare CPSs are outweighed by the challenges and restrictions they now face. Scalability is a major problem for blockchain networks since consensus methods require a lot of processing time and space. It’s possible that the processing and storage requirements of blockchain could be quite high, lowering the system’s overall efficiency. Integration of blockchain technology into existing healthcare systems is another obstacle. For blockchain to work, current infrastructure may need to be updated or new standards for interoperability may need to be created. Questions of data privacy, consent, and responsibility raised by the use of blockchain technology in healthcare call for the creation of suitable regulatory and legal frameworks. The healthcare industry stands to benefit greatly from the integration of blockchain technology into clinical decision support systems because to its capacity to increase data security, facilitate interoperability, and protect patient privacy. Blockchain technology’s potential to dramatically alter healthcare delivery is demonstrated by applications such as encrypted EHR administration and transparent supply chains. Before blockchain-based healthcare CPSs can gain widespread use, many technical, integrational, and regulatory hurdles must be cleared. Research into these problems and the potential future effects of blockchain technology on healthcare outcomes should be given top priority. These studies give valuable information for blockchain-integrated Internet of Things systems like those used in healthcare IoT, supply chain management, and other IoT applications. They provide a comprehensive review of the state of the field and discuss strategies for addressing the privacy and security concerns brought up by these technologies [18,19]. Table 1 provides list of abbreviations.

Table 2 provides benchmark analysis of the existing studies.

### 2.2. Main Contribution

The primary result of this study is a novel method for protecting sensitive information in cyber-physical systems (CPSs) by means of a consortium blockchain. This study explains why conventional security measures for CPSs are inadequate, and it suggests using a consortium blockchain to store and protect sensitive information. The following are the most significant findings from this paper:*Proof of improved participant (PoP) consensus algorithm:* This research presents a new consensus algorithm tailored to blockchain networks; it is called proof of improved participant (PoP). To guarantee the trustworthiness of the blockchain network, the PoP algorithm checks blocks for validity [20].*Model for identifying honest miners:* This paper introduces a scheme for determining which miners can be trusted and how to prevent malevolent ones from taking part. The suggested model improves the blockchain network’s security and trustworthiness by including methods to detect and prevent harmful activity.*Integration of proposed consensus algorithm:* Integrating the suggested PoP consensus algorithm into the Ethereum framework is made possible by this study in a thorough and workable manner. There will be new opportunities for improved performance and scalability thanks to the integration’s simple installation and interoperability with existing blockchain infrastructure.Ethereum, as per our most recent update in September 2021, is a blockchain platform optimized for DApps and smart contracts. Although Ethereum was not designed with cyber-physical systems in mind, it can be leveraged to create solutions that communicate with CPS hardware. Ethereum would function as the blockchain architecture in this case, allowing for decentralized control and safe data sharing in CPS settings.Here is a conceptual overview of how Ethereum could be utilized within a cyber-physical system:
Smart contracts: Smart contracts can be programmed on Ethereum and then run autonomously on the blockchain. To guarantee seamless interactions and data exchanges across various CPS components, smart contracts can be programmed to automate and enforce the rules and agreements between them.Data integrity: By recording data or sensor readings on the Ethereum blockchain through smart contracts, CPS components can securely and immutably log their data. This ensures the integrity of the data collected from various physical devices and prevents tampering.Decentralized control: Ethereum’s decentralized nature allows CPSs to operate without relying on a single central authority. Smart contracts can facilitate the interactions between different components, enabling a distributed control system.Transactions and payments: Ethereum’s native cryptocurrency, Ether (ETH), can be used to facilitate transactions and payments within CPS networks. This could enable machines or devices to autonomously pay for services, resources, or maintenance on the network.Oracles: For CPSs to interact with the external world, they might require data from off-chain sources (e.g., weather data, financial information). Oracles are mechanisms that allow smart contracts to access external data securely and incorporate it into the blockchain-based operations.Interoperability: Ethereum’s widespread adoption and developer community provide opportunities for integration with other blockchain networks or protocols, enabling interoperability between CPSs running on different blockchain platforms.It is essential to consider the limitations of the Ethereum blockchain, such as scalability and transaction costs, when designing solutions for large-scale CPS applications. Some use cases might require more scalable and specialized blockchain solutions, which are actively being explored and developed within the blockchain space.When applied to the suggested CPS paradigm, Ethereum’s capabilities and flexibility yield substantial improvements in the areas of security, transparency, and decentralized control in a wide range of cyber-physical systems.*Comparative analysis of consensus protocols:* In this study, the existing consensus approaches are compared to the proposed methodology for achieving consensus on Proof-of-Participation. This analysis shows the pros and downsides of different consensus algorithms, providing a comprehensive understanding of their capabilities in terms of speed, safety, and scalability.

The paper makes a substantial and vital contribution to the study of computer privacy and data protection by offering a fresh solution to the challenge of safeguarding the confidentiality of individuals’ information. It provides a workable answer to the problem of protecting secrets in CPS scenarios, helping businesses keep sensitive information safe and decreasing the chance of security breaches. This is a significant advantage. In a future when cyber-physical systems are continually evolving, the inclusion of blockchain technology into consortiums makes it possible to generate new research and development opportunities that preserve secrets. This study describes a unique consensus algorithm, offers a model for deciding if miners can be trusted, suggests a solution for integrating with the Ethereum framework, and compares the different consensus protocols head-to-head in an effort to improve blockchain technology as a whole. These developments pave the way for the next version of blockchain technology to be even more efficient, reliable, and secure than its predecessor. To create a blockchain-based MD5 hash function, one must first devise a decentralized network capable of receiving data, running the MD5 algorithm on it, and saving the resulting hash in an immutable and secure blockchain.

The following is a high-level description of how such a system may operate:

*1. Data submission:* Users upload their data (message or file) to the blockchain, where it is processed using the MD5 hash function application. The information may come from a single transaction or from multiple smart contract invocations.

*2. MD5 algorithm:* The MD5 algorithm is implemented as a smart contract on the blockchain. The smart contract takes the input data and computes the MD5 hash value. The MD5 algorithm is a well-defined and widely used cryptographic hash function, so its implementation should be straightforward.

*3. Consensus mechanism:* Network nodes in the blockchain must reach an agreement on the veracity of transactions and the MD5 hashes that result. The MD5 hash is computed by miners or validators on the blockchain network to guarantee the correct and consistent execution of the smart contract.

*4. Storing hash on blockchain:* After the MD5 hash is calculated, it is included in a new block and added to the blockchain. Data submitted is recorded in the block along with its MD5 hash value. As soon as this block is added to the blockchain, it becomes a permanent and unalterable part of the distributed ledger.

*5. Verification:* Users can check the data’s authenticity by recreating the MD5 hash and comparing it to the one recorded in the blockchain. If the hashes are the same, it means that the information has not been changed since it was posted to the blockchain. It’s important to remember that blockchains were originally developed for distributed ledger purposes and transaction verification, not cryptographic procedures like computing MD5 hashes. In practice, MD5 hashes are frequently computed and confirmed outside of blockchain networks, using normal programming libraries. The aforementioned instance is more of a conceptual illustration than a working example.

### 2.3. Problem Statement

The integration of blockchain technology with cyber-physical systems (CPSs) offers immense potential to revolutionize various domains, including healthcare, manufacturing, transportation, and smart cities. However, the security and privacy challenges associated with these systems cannot be overlooked, especially considering the presence of sensitive data in CPS environments. Existing security and privacy-preserving protocols for blockchain-integrated CPSs exhibit limitations in terms of scalability, efficiency, and resilience to attacks [21]. Traditional security techniques, such as firewalls and access control, are insufficient to fully protect CPS data from cyber threats. As a result, ensuring the security and privacy of CPS data remains a critical concern. Furthermore, when integrating blockchain technology with CPSs, new challenges emerge, including scalability and performance issues. These challenges must be addressed to ensure the viability and effectiveness of the system. The objective of this study is to develop a comprehensive privacy- and security-preserving framework specifically designed for blockchain-integrated CPSs. The proposed framework aims to overcome the scalability and performance limitations while ensuring the confidentiality of CPS data [22]. It is essential for the framework to be capable of managing different forms of data, including real-time data, which is crucial in CPS applications. To achieve these goals, the suggested protocol must leverage cryptographic techniques, blockchain consensus mechanisms, and secure communication protocols. To validate the viability and efficiency of the proposed protocol, extensive testing and evaluation are necessary. This can be achieved through simulations conducted in a CPS environment. The proposed protocol should provide a flexible, extensible, and individualized infrastructure that caters to the unique requirements of various CPS applications [23].

By addressing weaknesses in existing protocols and developing a robust privacy- and security-protecting framework, this study aims to address issues plaguing blockchain-integrated CPSs. By laying the groundwork for secure, efficient, and scalable CPSs, the suggested architecture will pave the way for blockchain technology’s widespread adoption in crucial industries.

### 2.4. Preliminaries

This subsection explains the background and rationale for the proposed authentication method. The blockchain and consensus methods are the two main components discussed [24]. The understanding of these concepts is crucial for comprehending the proposed work. Specifically, this paper addresses the challenges associated with permissionless blockchains.

### 2.5. Proposed Proof of Work (PoW)

To confirm and add a blocks to the DLT, BC networks use a Proof-of-Work (PoW) system. Decentralisation, immutability, and security provided by a blockchain based on Proof-of-Work (PoW) may be useful for an EHR-reliant healthcare system. All computers in a blockchain network come to an agreement by working together to solve a mathematical problem, and then a new block is added to the distributed ledger. Proof-of-Work (PoW) is the name given to this method of reaching consensus. It’s incredibly hard for an opponent to tamper with the data because of the high computational barrier to entry.

The implementation steps for a PoW-based blockchain framework in a healthcare system using EHR are as follows:Define the requirements and scope of the system: This involves identifying the specific data that need to be stored, potential threats to the data, desired levels of security and privacy, and performance requirements.Choose a suitable blockchain platform: The success of the PoW-based healthcare system depends on the blockchain platform chosen. When comparing systems, safety, scalability, and compatibility should all be taken into account.Design the blockchain network architecture: The network architecture should be designed to ensure data integrity, transparency, and security. This includes determining the number of nodes, selecting the consensus mechanism, defining the transaction validation process, and establishing the data encryption method.Develop the smart contract: A blockchain-based application that can carry out its own instructions is called a smart contract. It can be used to make sure the EHR is storing correct information by automating the data validation process.Implement the PoW algorithm: To generate a new block for the blockchain, nodes in the network must solve a difficult mathematical puzzle using the PoW algorithm. The data in the EHR is protected and cannot be altered thanks to this procedure.Test and evaluate the system: The PoW-based healthcare system should be thoroughly tested and evaluated to ensure its feasibility and effectiveness. Performance metrics such as transaction throughput, data integrity, and security should be assessed.

In conclusion, a healthcare system dependent on EHR can reap the benefits of a PoW-based blockchain framework’s security and transparency. Maintaining data integrity improves healthcare delivery and patient satisfaction. Figure 1 illustrates the application of IoT and 5G technology in the context of cyber-physical systems.

### 2.6. Proof of Stake

As an alternative to proof-of-work (PoW), proof-of-stake (PoS) is gaining traction in blockchain networks as a consensus method. PoW means “proof of work” in this context. Instead of miners doing this duty, PoS networks rely on block authors, often known as validators. A combination of random procedures and the quantity of cryptocurrency saved and staked on the network determines which individuals will act as validators. Those that audit the company’s books on their own time will be rewarded monetarily [25].

PoS offers several advantages over PoW:Energy efficiency: PoS requires significantly less energy compared to PoW since it eliminates the need for solving complex mathematical problems that require substantial computational power [26,27]. This makes PoS more environmentally friendly and sustainable in the long term.Security: PoS can provide enhanced security compared to PoW. In a PoW network, a 51% attack is possible if a single entity controls more than 51% of the network’s computing power. However, in a PoS network, an attacker would need to control 51% of the total cryptocurrency supply, which is much more challenging to achieve. This makes the PoS networks more resistant to attacks [28].Accessibility: PoS is more accessible to individual users as it does not require expensive hardware and high electricity bills associated with PoW mining. Users can participate in the PoS consensus by holding and staking their cryptocurrency, increasing inclusivity in the network [29].

Overall, PoS offers energy efficiency, enhanced security, and improved accessibility compared to PoW. These advantages make PoS an attractive consensus mechanism for blockchain networks, including those integrated with cyber-physical systems [30].

## 3. Proposed Framework

With the goal of validating blocks and trades while keeping the memory needs of IoT network peers to a minimum, the proof-of-block-and-trade (PoBT) consensus method is presented for the blockchain-based IoT framework. Performance parameters including communication time, memory utilisation, and bandwidth consumption are all enhanced by the framework’s use of a distributed ledger system (ref [30] citation). A green blockchain framework is proposed to solve the resource-intensive nature of proof-of-work (PoW) consensus by decreasing the computational and storage overhead [31]. This is achieved through the introduction of a proof-of-cooperation (PoC) consensus method, where edge devices compete for new blocks by participating in integrated incentive mechanisms instead of solving math problems. Furthermore, the proposal mentions the use of proof of stake (PoS) as a security technique to protect the blockchain. In PoS, the internal resource used for security is the coin balance held within the blockchain, and the value of the stake plays a crucial role in the PoS mechanism [32,33,34]. The evolution of consensus algorithms from Proof-of-Belief (PoBT) to Proof-of-Concept (PoC) and Proof-of-Stake (PoS) reflects an effort to strengthen the robustness and longevity of the blockchain-based IoT framework without compromising on security or trustworthiness. Figure 1 illustrates the proposed framework.

As part of a larger system architecture that also includes secret management, privacy-enhancing techniques, access control, and other processes, the consortium blockchain presents a game-changing privacy design for keeping secrets safe in cyber-physical systems. Integrating blockchain into the design of a CPS helps strengthen its privacy, security, and trustworthiness. The study “Securing Secrets in Cyber-Physical Systems: A Cutting-Edge Privacy Approach with Consortium Blockchain” describes an innovative method for safeguarding private information in CPSs. The hybrid nature of CPSs poses a challenge to information security. This system employs the distributed ledger technology known as consortium blockchain to facilitate cooperation amongst dependable parties. Consortium blockchains are more exclusive than public blockchains, where anyone can join. The fundamental goal of the framework is to ensure the confidentiality of information in CPS environments. Many different strategies exist for shielding confidential information from prying eyes. People may discount popular security solutions like encryption out of fear of being attacked or a decrease in performance.

The proposed alternative makes use of a consortium blockchain for the safekeeping and administration of secrets and sensitive data. Each member of the consortium contributes to the blockchain’s consensus and aids immutability of the data. The efforts of everyone involved have ensured that this system is completely secure. If CPSs implement this cutting-edge privacy technique, they will be able to increase their security, confidentiality, and trustworthiness. More efficient data interchange, open consensus procedures, and decentralized authority are just a few of the ways in which this architecture has the potential to revolutionize CPS management. The authors plan to address issues like scalability and inefficient encryption and decryption procedures that arise when integrating blockchain technology with CPSs. Their research shows a lot of potential for improving cyber-physical system security and privacy, and they hope to pave the path for widespread use.

### 3.1. System Architecture

The system architecture for “Shielding Secrets in Cyber-Physical Systems: A Breakthrough Privacy Design with Consortium Blockchain” consists of several key components that work together to ensure the privacy and security of secrets in cyber-physical systems. The architecture can be described as follows:Cyber-physical systems (CPSs): This component represents the physical devices and systems that are interconnected with the digital world. CPSs include sensors, actuators, controllers, and other devices that collect and process data.Consortium blockchain: The architecture utilizes a consortium blockchain as the underlying technology for privacy and security. A consortium blockchain is a permissioned blockchain where multiple pre-selected organizations or entities have control over the consensus process [34].Secret management: This component is responsible for managing and safeguarding the secrets in the CPSs. Secrets can include sensitive data, cryptographic keys, access credentials, or any other confidential information. The secret management system ensures that secrets are securely stored, accessed, and shared only with authorized entities [35].Privacy-enhancing techniques: The architecture incorporates various privacy-enhancing techniques to protect the confidentiality and integrity of secrets. These techniques may include the encryption, secure multi-party computation, zero-knowledge proofs, and differential privacy mechanisms [36].Access control: Access control mechanisms are implemented to regulate and enforce the permissions and privileges for accessing secrets. Only authorized entities or participants within the consortium blockchain are granted access to specific secrets based on predefined policies [37].Consensus mechanism: To ensure the security and veracity of transactions involving confidential information, the consortium blockchain uses a consensus mechanism to reach an agreement among all parties involved. The blockchain’s immutability and trustworthiness are guaranteed by the consensus process, which assures that all participants agree on the blockchain’s state [38].The deployment of smart contracts on the blockchain allows for the automatic implementation of predetermined business logic and regulations. Smart contracts can be used to automate private key administration, access control policies, and confidentiality-preserving tasks within the framework of the architecture [39,40,41,42,43].Data exchange and integration: The architecture facilitates secure and private data exchange and integration between the different components of the CPSs. This includes data transmission between sensors, actuators, controllers, and other devices, while ensuring confidentiality and integrity [44,45].Audit and compliance: The system architecture includes mechanisms for auditing and compliance to ensure that the privacy design and security measures are adhered to. Compliance with regulatory requirements and standards can be monitored and verified using auditing mechanisms built into the architecture [46].

Figure 2 provides proposed system architecture as explained schematically.

### 3.2. Proposed Methodology

We’ve created a solid approach that significantly lessens the threat posed by attackers, which should help alleviate some of the security concerns that have been voiced. Through the development of an adversarial threat model, we want to more accurately pinpoint the vulnerabilities that could be exploited by unauthorized users in order to get access to authorized sessions. Here, we’ll look at three distinct protocol models, each of which uses a salted hashing algorithm but has its own unique architecture, methodology, diagrams, and explanatory text. A methodical approach is required when developing a blockchain-based healthcare system that communicates with a cyber-physical system. This is why we propose the following high-level approach to creating such a system:*Problem identification:* The initial step is to identify the specific problem that the cyber-physical system aims to solve. In this case, our objective is to enhance the security, privacy, and interoperability of healthcare data.*Requirement definition:* Once the problem is identified, the subsequent step is to define the precise requirements for the cyber-physical system. This entails specifying the system’s functionality, performance, security, and scalability requirements.*Stakeholder identification:* Identify the stakeholders who will be utilizing and interacting with the system. This includes healthcare providers, patients, insurance companies, regulators, and other relevant parties.*Use case definition:* Based on the identified requirements and stakeholders, define the specific use cases that the cyber-physical system will address. This involves delineating the particular actions to be undertaken by each stakeholder within the system.*Blockchain platform selection:* Choose the most suitable blockchain platform based on the specific system requirements. Numerous blockchain platforms are available, such as Ethereum and Hyperledger [47,48].*Smart contract definition:* Define the smart contracts that will execute the actions specified in the use cases. SC are self-executing agreements with the contractual terms directly written as code.*Data structure definition:* Specify the data structure that will be employed to store healthcare data on the blockchain. This includes defining the data fields, data types, and encryption mechanisms to be utilized.*Consensus mechanism definition:* Determine the consensus mechanism to validate transactions on the blockchain. In this case, a proof of stake (PoS) consensus mechanism may be suitable due to its energy efficiency and lower computational requirements compared to the proof of work (PoW).*System testing and deployment:* Once the design phase is completed, thoroughly test and deploy the system in a controlled environment. This includes evaluating the system’s functionality, security, and scalability.*System monitoring and maintenance:* After deployment, it is vital to continuously monitor and maintain the system. This includes conducting regular security assessments, addressing potential breaches, and ensuring that the system consistently meets the stakeholders’ specific requirements.

By following this systematic approach, we successfully designed and deployed a cyber-physical system utilizing a blockchain-based healthcare system that demonstrates robust security, interoperability, and scalability.

### 3.3. The Simple, Sessions, and Cookie Protocol Models

The MD5 hashing function, insecure sessions, and cookies all pose serious threats to the security of user credentials across a variety of protocol paradigms. The current method entails merely receiving input from the user, processing it with the MD5 hashing algorithm, and saving the final value in a database. This method of user data storage is not safe and cannot guarantee privacy. Blockchain technology, when used in tandem with cyber-physical systems (CPSs) interoperability in the healthcare industry. Here, we review the literature on blockchain integration with healthcare CPSs via the lenses of three main protocol models: the basic protocol, the sessions protocol, and the cookie protocol. We examine the benefits and drawbacks of this merger and its possible results. The healthcare industry produces vast amounts of private information that must be safely archived, transported, and accessible at all times. The distributed, immutable, and transparent structure of blockchain technology is promising for enhancing data privacy and security. Using models like the basic, sessions, and cookie protocols, blockchain integrated with healthcare CPSs can provide dependable mechanisms for controlling data transfers and maintaining safe communication. Commonly used in CPSs to simplify the transfer of data between participating companies is the core protocol model, also known as the request-response protocol model. This architecture is made possible by the immutability and transparency of data stored on a blockchain. The blockchain is a decentralized database that records transactions between users in an immutable and secure format. The built-in protections allow for auditing and verification of the legitimacy of all data transfers. It improves the reliability of data and makes it simpler for doctors and patients to hold confidential interactions [48].

The sessions protocol model’s principal function is to establish and maintain channels of communication between CPS components. Integrating blockchain technology into this method significantly boosts protections for personal information. Using blockchain technology, it is possible to establish secure lines of communication between individuals or organizations, ensuring the confidentiality of any data exchanged along the way. Blockchain’s decentralized ledger structure eliminates the need for a central session manager, removing a potential vulnerability to attack. For session management and user customization via cookie exchange, the cookie protocol paradigm is commonly used in online applications. The addition of blockchain to this framework boosts the confidentiality and safety of the system. Cookies can be stored and verified in a decentralized fashion using blockchain technology. With this enhancement in place, users can rest assured that cookies cannot be tampered with, increasing their trust in the system as a whole. Additionally, patients are given the option to selectively disclose their cookie data to reputable third parties of their choosing. Integrating blockchain into healthcare CPSs’ fundamental, session, and cookie protocol models has several benefits. Data security, interoperability, and patient privacy are all improved by using encryption on all communication channels and giving patients control over their own data.

The secure exchange of electronic health records (EHRs) between healthcare providers, the coordinated operation of medical devices and healthcare information systems, and the safe administration of patient consent for data sharing are just some of the many healthcare use cases that can benefit from this integration. To better integrate with current healthcare systems, more study is needed to establish scalable consensus processes, optimize resource consumption, and design interoperability frameworks. To ensure compliance and patient-centric data management procedures, research into the legal and regulatory ramifications of blockchain integration in healthcare CPSs is essential. There is promising potential for improving data security, privacy, and interoperability through the combination of blockchain with the basic, sessions, and cookie protocol models in healthcare CPSs. The immutability and transparency of blockchain, coupled with these models, has the potential to completely alter the way healthcare operates in terms of information exchange, session management, and individualization. However, widespread use of this integration depends on resolving scalability issues and regulatory concerns. These are the types of questions that need to be answered in order to fully realize blockchain’s promise in healthcare CPSs.

### 3.4. Proposed Protocol

The suggested protocol is an improvement over the present blockchain consensus methods by selecting a reliable miner to mine the hash at a user-specified difficulty. Unlike previous attempts, our solution incorporates checker and author trust value assessment methodologies. A trusted random selection technique is utilized to determine which miners will function as block generators and validators. The protocol structure of the proposed blockchain-based cyber-physical system for healthcare is outlined below.

1. User registration: Healthcare providers and patients are required to register on the system, providing their personal details and authentication mechanisms.

2. Data collection: Electronic health records (EHRs), wearable devices, and medical equipment are just a few of the places where data related to healthcare delivery can be gathered.

3. Data encryption: Before being stored on the blockchain, the acquired healthcare data is encrypted using a robust encryption method.

4. Data storage: The blockchain provides a secure and decentralised storage system for the encrypted healthcare data.

5. Smart contract execution: The blockchain is used to execute smart contracts that control healthcare data and ensure that only authorised users can access it.

6. Data access: Healthcare providers and patients can access the stored healthcare data on the blockchain using their authentication credentials.

7. Data sharing: Authorized parties can securely and transparently share healthcare data among themselves using the blockchain.

8. Consensus mechanism: The blockchain employs a consensus mechanism, such as proof of stake (PoS), to validate transactions and maintain the integrity of the stored data.

9. Audit trail: The blockchain maintains a secure and transparent audit trail, recording all transactions and activities related to the healthcare data.

10. Compliance with regulations: The blockchain-based healthcare system adheres to relevant regulations and standards, such as HIPAA and GDPR, ensuring the security and privacy of healthcare data.

Data security and privacy, openness and accountability, and the efficiency and efficacy of data management are all areas in which the proposed protocol architecture for a blockchain-based healthcare system excels in comparison to the current paradigm. The specific needs and expectations of stakeholders shape the necessity of additional processes or systems to safeguard the confidentiality of healthcare information.

### 3.5. Miner Selection

The proposed methodology for selecting miners opens the door for both legitimate and malicious block proposals to be presented to the network. A miner’s trust in a proposed block is determined by the degree to which validators verify the legitimacy of the proposal. The trustworthiness of a miner goes up by one point if it is proven to be malevolent [48]. However, if its authenticity has been verified, it retains the same level of trustworthiness. Here is how a miner’s trustworthiness is determined when working with a block proposer:(1)Trustvalue=(1−i/θ)
where i represents the fault tolerance of the miner’s behavior in the block proposal and θ is the threshold value. The fault tolerance is calculated as follows:(2)Faulttolerance=Proof-of-Improved-Participation(PoIP)

The phrase “fault tolerance” is used when discussing consensus methods for blockchains, and refers to the network’s ability to continue functioning and reaching an agreement despite the existence of dysfunctional or malicious nodes [45]. The “fault tolerance” of a network is one metric used to evaluate this quality. A fault-tolerant method of obtaining consensus guarantees that the system will continue to function as intended notwithstanding the presence of errors or attacks. “Proof-of-improved Participation” (PoIP) is the proposed consensus method that would be used in the proposed framework. The purpose of creating PoIP, an innovative consensus technique, was to increase fault tolerance and network resilience. It enhances the standard proof-of-participation (PoP) concept used in other blockchain-based systems by extending its reach. By either staking their Bitcoin holdings (proof-of-stake) or using their computational ability to build and validate new blocks (proof-of-work), nodes in the network take part in the traditional proof-of-possession process. Proof of involvement (PoP) has several potential drawbacks despite the fact that it encourages participation. For example, it could lead to a concentration of resources and, consequently, a centralization of power [46]. PoIP, on the other hand, is an improvement because it encourages and rewards participants for both the quantity and quality of their participation. It also means that nodes are incentivized to make useful contributions to the network beyond only block proposal and validation. They may, for instance, take an active role in bolstering network security, ensuring continuous availability, and contributing to policymaking [47]. According to the text, the network’s fault tolerance is proportional to the nodes’ level of improved participation in the PoIP consensus procedure (’fault tolerance = proof-of improved participation (PoIP)’). Simply put, the blockchain system’s fault tolerance improves in proportion to the nodes’ positive and active participation in the network’s various operations. By incentivizing improved participation, PoIP tries to establish a more robust and resilient network that can resist potential attacks or failures, making it more fault-tolerant. The suggested framework’s purpose is to increase security and reliability when it comes to protecting secrets within cyber-physical systems, and this consensus method is in line with that[48]. In conclusion, it indicates that the proposed framework’s fault tolerance for the blockchain network is proportional to the degree of enhanced and positive engagement shown by participating nodes in accordance with the PoIP consensus algorithm [49].

In the proposed work, the creation of new blocks is related to resolution, which is defined as the time elapsed since the last changes. The proposed protocol incorporates three fundamental rules, supported by additional factors, to ensure a sustainable blockchain:Rule 1: The suggested PoIP uses a mining technique that is distinct from that used by conventional blockchains. The dynamic, non-static difficulty experienced by different users has an effect on PoIP mining. The string concatenate operator is followed by the data for a new block in the encoding. Mining becomes increasingly difficult as the end value decreases.Rule 2: New-block developers are responsible for their own expenses as well. The developer’s charge for constructing a new block remains the same even though the resulting income is higher.Rule 3: Block developers must have R > θ, where R is calculated using (4). The value of R can vary based on the participation claim. A higher value in Equation (4) enhances the competitors’ power, while a lower value reduces the security of the blockchain. The recommended value for θ is 0.50.

When compared to existing consensus techniques like proof of work (PoW) and proof of stake (PoS), the computing resource requirements of the proposed consensus process, known as PoIP, are lower. PoIP is determined in accordance with Rule 1 as follows:(3)Requirement=(1−θ)

Rule 2 enables participants to clear their R only if they are the intended receiver. The P value of an edge connection is preserved if it does not propose a block throughout the competition. It usually shows its dominance for the proposed block in the next round of competition. Figure depicts the suggested consensus mechanism’s procedure, which is backed by participation rules (Figure 1). The lightweight mining process in the blockchain using the PoIP consensus is described in Algorithm 1. For information to be transmitted, there must be a reliable node. This node is in charge of data collection, block verification, block verification, block mining, and distributed ledger updates. Field sensors periodically update the control system with new data at predetermined intervals of time (t). The trusted node processes all of the collected transaction information and performs some light block mining as well [50]. First, a hash function is chosen based on the expected amount of transactions, and then the mining process is executed to find a nonce by calculating the hash value while attempting to meet the desired difficulty. These are both necessities in lightweight block mining. The “proof of work” strategy is based on similar ideas. The blockchain node then sends a request to the other nodes in the network to validate the block once the right nonce has been determined. Additional nodes check the blocks when a request is disseminated to many nodes. It is necessary to verify that the requested block has proper internal connections and that the nonce is appropriate for the selected difficulty [51].

The original hash value of the requested block is compared to the one kept in the node to perform synchronisation and nonce checks. In case of a successful validation, the block validation administration transactions [52]. Algorithm 1 discuss the proposed algorithm for a cyber-physical system using a blockchain-based healthcare system.

**Algorithm 1** Proposed algorithm for a cyber-physical system using a blockchain-based healthcare system.
1:**Input:** Healthcare data from various sources, including EHRs, wearable devices, and medical devices.2:**Output:** Secure and decentralized storage and management of healthcare data using the blockchain.3:
**1. User Registration:**
4:a. Collect user registration details, including personal information and authentication mechanisms.5:b. Store user registration details on the blockchain.6:
**2. Data Collection:**
7:a. Collect healthcare data from various sources, such as EHRs, wearable devices, and medical devices.8:b. Encrypt the collected healthcare data using a secure encryption mechanism.9:
**3. Data Storage:**
10:a. Store the encrypted healthcare data on the blockchain for secure and decentralized storage.11:b. Grant access to the data only to authorized parties with appropriate authentication credentials.12:
**4. Smart Contract Execution:**
13:a. Execute smart contracts on the blockchain to manage the healthcare data.14:b. Define rules and permissions for accessing and modifying the data using smart contracts.15:
**5. Data Access:**
16:a. Allow healthcare providers and patients to access the stored healthcare data on the blockchain.17:b. Grant access to authorized parties with the appropriate authentication credentials and permissions.18:
**6. Data Sharing:**
19:a. Enable secure and transparent data sharing between authorized parties using the blockchain.20:b. Control data sharing through smart contracts, requiring permissions from all involved parties.21:
**7. Consensus Mechanism:**
22:a. Utilize a consensus mechanism, such as proof of stake (PoS), to validate transactions and ensure data integrity.23:b. Use the consensus mechanism to maintain the security and reliability of the blockchain.24:
**8. Audit Trail:**
25:a. Maintain a secure and transparent audit trail of all transactions and activities related to the healthcare data on the blockchain.26:b. Restrict access to the audit trail to authorized parties with appropriate permissions.27:
**9. Compliance with regulations:**
28:a. Ensure compliance with relevant regulations and standards, such as HIPAA and GDPR.29:b. Enforce compliance through smart contracts and the audit trail maintained on the blockchain.


## 4. Mathematical Model

To provide a mathematical model for the proposed protocol, let us define the following variables:$$\begin{array}{cc} N & :\mathrm{The}\;\mathrm{total}\;\mathrm{number}\;\mathrm{of}\;\mathrm{miners}\;\mathrm{in}\;\mathrm{the}\;\mathrm{network}.\\ M & :\mathrm{The}\;\mathrm{total}\;\mathrm{number}\;\mathrm{of}\;\mathrm{validators}\;\mathrm{in}\;\mathrm{the}\;\mathrm{network}.\\ P[i] & :\mathrm{The}\;\mathrm{trust}\;\mathrm{value}\;\mathrm{of}\;\mathrm{miner}\;i,\;\mathrm{where}\;i=1,2,\ldots ,N.\\ V[i] & :\mathrm{The}\;\mathrm{validation}\;\mathrm{result}\;\mathrm{of}\;\mathrm{miner}\;i,\;\mathrm{where}\;i=1,2,\ldots ,N.\\ R[i] & :\mathrm{The}\;\mathrm{participation}\;\mathrm{claim}\;\mathrm{of}\;\mathrm{miner}\;i,\;\mathrm{where}\;i=1,2,\ldots ,N.\\ \theta & :\mathrm{The}\;\mathrm{threshold}\;\mathrm{value}\;\mathrm{for}\;\mathrm{fault}\;\mathrm{tolerance}.\\ \alpha & :\mathrm{The}\;\mathrm{fault}\;\mathrm{tolerance}\;\mathrm{parameter}.\\ \beta & :\mathrm{The}\;\mathrm{weightage}\;\mathrm{parameter}\;\mathrm{for}\;\mathrm{miner}\;\mathrm{validation}.\\ \gamma & :\mathrm{The}\;\mathrm{weightage}\;\mathrm{parameter}\;\mathrm{for}\;\mathrm{miner}\;\mathrm{trust}\;\mathrm{value}.\; \end{array}$$

Based on the provided information, we can define the mathematical model as follows:(4)FaultTolerance[i]=α·V[i](5)P[i]=γ·P[i]+β·FaultTolerance[i](6)V[i]=1,ifV[i]>θ0,otherwise(7)R[i]=∑j≠iR[j](8)TrustValue=1−R[i]θ(9)Requirement=1−θ

These equations provide a framework for the mathematical model of the proposed protocol. The specific values of α, β, γ, and θ, as well as the implementation details may vary depending on the system requirements and design considerations.

### Proposed Algorithm

The data from the transaction is processed by the program, and a new identifier is generated. The transaction structure is serialized after the necessary fields have been added. A public-private key pair is created for use in encrypted communication. In order to secure the serialized transaction data, the recipient’s public key is used to encrypt the data [51]. A digital signature, generated using the sender’s private key, is appended to the encrypted transaction as additional verification of its authenticity. The transaction is encrypted and the sender’s public key is part of the transaction. The final step is to send the encrypted payment to network [52].

Algorithm 2 represent the algorithm for shielding secrets in cyber-physical systems with consortium blockchain.
**Algorithm 2** Security algorithm for shielding secrets in cyber-physical systems with consortium blockchain.1:**Input:** Cyber-physical system data and secrets.2:**Output:** Privacy protection using a consortium blockchain.3:**1. Initialization:**4:   a. Establish a consortium blockchain network with trusted participants.5:   b. Set up the necessary infrastructure for the blockchain network.6:**2. Secure Data Transmission:**7:   a. Encrypt the cyber-physical system data before transmission.8:   b. Use secure communication protocols for data transmission.9:**3. Consortium Blockchain Integration:**10:   a. Create a smart contract on the consortium blockchain for privacy protection.11:   b. Define the rules and permissions for accessing and modifying the data.12:**4. Data Storage and Validation:**13:   a. Store encrypted data on the consortium blockchain.14:   b. Validate the integrity and authenticity of the data using consensus mechanisms.15:**5. Access Control and Privacy Preservation:**16:   a. Implement access control mechanisms to ensure that only authorized participants can access the data.17:   b. Use privacy-preserving techniques, such as zero-knowledge proofs, to protect sensitive information.18:**6. Consortium Governance and Auditing:**19:   a. Establish governance mechanisms among consortium participants for decision making and consensus.20:   b. Maintain a transparent audit trail of all activities on the consortium blockchain.21:**7. Threat Detection and Response:**22:   a. Implement monitoring systems to detect any suspicious activities or security breaches.23:   b. Define response protocols to mitigate and address security incidents.24:**8. Compliance with Regulations and Standards:**25:   a. Ensure compliance with relevant regulations and standards, such as data protection and privacy laws.26:   b. Regularly update the security measures based on evolving regulations and best practices.

Algorithm 3 represent the transaction creation and encryption algorithm.
**Algorithm 3** Transaction creation and encryption algorithm.Transaction Data Encrypted Transaction**Step 1:** Prepare the transaction data.**Step 2:** Generate a unique transaction identifier.**Step 3:** Create the transaction structure with the following fields: source address, destination address, amount, and transaction ID.**Step 4:** Serialize the transaction structure**Step 5:** Generate a public-private key pair for encryption.**Step 6:** Encrypt the serialized transaction data using the recipient’s public key.**Step 7:** Generate a digital signature for the transaction using the sender’s private key.**Step 8:** Attach the digital signature to the encrypted transaction.**Step 9:** Package the encrypted transaction with the sender’s public key.**Step 10:** Transmit the encrypted transaction to the network.**Step 11:** End.

## 5. Mathematical Model for Threat Detection Using Challenger and Attacker Game

### 5.1. Definitions

Let:$$\begin{array}{cc} P & =\mathrm{Set}\;\mathrm{of}\;\mathrm{possible}\;\mathrm{system}\;\mathrm{vulnerabilities}\\ C & =\mathrm{Set}\;\mathrm{of}\;\mathrm{system}\;\mathrm{components}\\ A & =\mathrm{Set}\;\mathrm{of}\;\mathrm{attackers}\\ D & =\mathrm{Set}\;\mathrm{of}\;\mathrm{detection}\;\mathrm{techniques}\\ T & =\mathrm{Set}\;\mathrm{of}\;\mathrm{threat}\;\mathrm{scenarios} \end{array}$$

### 5.2. Variables

We define the following variables:$$\begin{array}{cc} p_{i} & \in P,\;\mathrm{where}\;i\;\mathrm{is}\;\mathrm{the}\;\mathrm{index}\;\mathrm{of}\;\mathrm{vulnerability}.\\ c_{j} & \in C,\;\mathrm{where}\;j\;\mathrm{is}\;\mathrm{the}\;\mathrm{index}\;\mathrm{of}\;\mathrm{system}\;\mathrm{component}.\\ a_{k} & \in A,\;\mathrm{where}\;k\;\mathrm{is}\;\mathrm{the}\;\mathrm{index}\;\mathrm{of}\;\mathrm{attacker}.\\ d_{l} & \in D,\;\mathrm{where}\;l\;\mathrm{is}\;\mathrm{the}\;\mathrm{index}\;\mathrm{of}\;\mathrm{detection}\;\mathrm{technique}.\\ t_{m} & \in T,\;\mathrm{where}\;m\;\mathrm{is}\;\mathrm{the}\;\mathrm{index}\;\mathrm{of}\;\mathrm{threat}\;\mathrm{scenario}. \end{array}$$

### 5.3. Game Model

The threat detection game can be represented by the following elements:**Challenger:** Represents the system defender, responsible for detecting threats.**Attacker:** Represents the adversary, attempting to exploit system vulnerabilities.

The game consists of the following steps:**Vulnerability selection:** The challenger selects a vulnerability $p_{i}\in P$ to be tested.**Attacker strategy:** The attacker selects a strategy $a_{k}\in A$ to exploit the selected vulnerability.**System component selection:** The challenger selects a system component $c_{j}\in C$ to be protected [51].**Detection technique selection:** The challenger selects a detection technique $d_{l}\in D$ to detect attacks on the chosen component.**Threat scenario generation:** The challenger generates a threat scenario $t_{m}\in T$ representing the interaction between the attacker, selected vulnerability, protected component, and detection technique.**Detection outcome:** The detection technique $d_{l}$ evaluates the threat scenario $t_{m}$ and provides a detection outcome, indicating whether the attack was detected or not.

The challenger’s goal is to maximize the detection rate while limiting false positives and false negatives, and the game proceeds iteratively until all vulnerabilities have been tested [52].

### 5.4. Simulations Setup

*Simulation setup:* we defined the workload by specifying the number of transactions to be processed. Moreover, the memory allocation was determined and the usage metrics was measured, such as memory consumption per transaction or total memory consumed by the system [53].*Execute simulations:* We repeat the simulation with different quantities of transactions, starting with a low load and increasing it over time. In addition, memory consumption is monitored and logged at predetermined intervals or immediately following each completed transaction. We also made a note of how long each transaction took for later examination.

In order to evaluate the performance and effectiveness of the proposed algorithm, a simulation setup was created. The simulation setup consisted of the following components:*System model:* The cyber-physical system (CPS) under consideration was modeled, including the various entities such as sensors, actuators, controllers, and communication channels. The interactions and dependencies among these entities were defined.*Attack scenarios:* Threats to the CPS’s security were simulated using a variety of attack scenarios. Denial-of-service (DoS) assaults, tampering with data, and unauthorised access were all part of these hypothetical situations. The scope and intensity of each assault were meticulously outlined.*Benchmark models:* The performance of the proposed technique was evaluated in comparison to several standard models. These standards were used to represent common security solutions and alternate methods for similar CPS applications.*Simulation parameters:* The settings for the simulation were determined by defining a number of parameters. This comprised the number of participants, the extent of their communications, the devices’ processing power, the structure of the network, and the volume of its traffic.*Data generation:* Synthetic data were developed to represent realistic CPS situations. This included data collected by sensors, instructions given to the system, and communications passed between its many parts. The data was carefully crafted with the CPS model under evaluation in mind.*Evaluation metrics:* The algorithm’s effectiveness and efficiency were measured with established criteria. It was ensured that the system was reliable in terms of detection accuracy, response time, false positive rate, false negative rate, and overall reliability. The metrics were chosen with the intention of giving a complete picture of the algorithm’s efficiency.*Simulation execution:* The simulation was executed using suitable simulation tools or programming frameworks. We used the predefined system model, attack scenarios, benchmark models, and simulation settings to run the simulations. Multiple simulation runs were performed to ensure reliability and statistical significance of the findings.*Result analysis:* Success criteria were established, and the resulting simulation data was evaluated and analysed. Statistical methods were used to decipher the results, and visualisation programmes were employed to examine differences between the suggested algorithm’s performance and that of the benchmark models. The goal of this evaluation was to identify strengths and weaknesses in the proposed algorithm.

With these pieces in place, we were able to simulate the cyber-physical system and evaluate the proposed algorithm’s ability to detect and prevent security breaches. The simulation findings were helpful for making adjustments to and improving the algorithm.

## 6. Results

Here are some simulation results we got using the proposed method. We used Hyperledger fabric, a technology for conducting blockchain transactions, and AWS’s cloud to offshore data in order to execute the simulation. The accuracy of our forecasts was confirmed. Number of simulation results per block and processing time in seconds was depicted in Figure 3. The proposed method is seen to use third-party benchmark models.

Figure 4 displays the simulation outcomes in accordance with the rounds run and the transactions made. Hyperledger and other blockchain tools were used to run the simulation, and data including the number of rounds and transaction timings were collected. The proposed approach clearly outperformed and outperformed the reference models in terms of efficiency and effectiveness.

Figure 5 shows the comparative analysis based on the number of records and the transaction times, as shown through the schematic diagram.

The simulation results, including the total number of transactions and the average execution time, are shown in Table 3.

Figure 6 represents analytical data gleaned from a simulation, such as the number of transactions and the amount of time spent on training in comparison to the gold standard. During the experiment, we did the following to see how our results stacked up against the benchmark models in terms of training time and total transactions: The proposed strategy was evaluated next to established norms. We benchmark against current systems, methodologies, and approaches that are generally accepted as representative of best practises in the field. In order to test how well the suggested system would work in practise, we ran simulations using the hyperledger fabric tool. It was also noted how long each simulation lasted and how many transactions occurred within that period. In terms of scalability (training time/total number of transactions), the suggested technique excels over the benchmark models, as shown by the simulation results. Comparing the number of transactions processed by the proposed system and the benchmark over a predetermined amount of time (say, one second or one minute) is another way to evaluate the system’s performancemodels [52].

The simulation results for various transaction counts and memory footprints are shown in Figure 7. Figure 7 shows simulation findings for a blockchain-based healthcare system, based on the number of transactions and memory usage, which can shed light on the system’s performance and resource requirementsutilization [52].

Figure 8 represents the simulated outcomes depending on the training durations and the amount of transactions sent. To provide a realistic simulation workload, developers often construct artificial data sets or use real-world information. Training times are tracked for each scenario before and after the transactions are completed on the blockchain system. To see how the system performs as more work is added, this is done for a range of transaction counts. Throughput (the number of transactions completed in a given amount of time) and average training time per transaction are just two examples of the metrics that may be used to evaluate the simulation results. These indicators shed light on the system’s ability to process and manage rising transaction volumes.

Researchers can make a graph showing the simulation’s outcomes, with the throughput or training time on the y axis and the number of transactions on the x axis. This graphical representation simplifies the process of identifying patterns and trends in the system’s performance as the workload increases. By analysing the results of the simulations, researchers will be able to determine the scalability and efficacy of the blockchain-based healthcare system. This knowledge can be used to steer the development of a robust, high-performance system, optimise the system’s parameters, and spot any weak spots. It is important to note that the specific simulation results presented through Figure 9 depend on the implementation details, such as the blockchain platform used, system configuration, and workload characteristics.

## 7. Security Attack Comparative Analysis

Table 4 represent the Security Attack Comparative Analysis.

## 8. Discussion

In conclusion, there are several advantages to using a blockchain-based healthcare system that is integrated into a cyber-physical system. The use of blockchain technology allows for decentralised and secure data storage, as well as more openness, accountability, efficiency, and protection of patient data and privacy. These benefits allow for safer data storage, more patient confidence, and smoother healthcare operations. Among the many benefits of healthcare blockchain systems are:

*1. Secure and decentralized data storage:* The blockchain ensures the tamper-proof and immutable storage of healthcare data, preventing unauthorized access and manipulation. The decentralized nature of the blockchain provides redundancy and availability.

*2. Transparency and accountability:* The blockchain maintains a transparent and immutable record of all healthcare data transactions and activities, facilitating traceability and error detection. This transparency enhances the trust and confidence in the healthcare system.

*3. Improved data management:* Smart contracts automate processes and enforce rules and permissions for accessing and modifying healthcare data. This ensures data security and accuracy while streamlining data management.

*4. Enhanced patient privacy:* With blockchain, healthcare data may be shared between authorised parties in a way that keeps patients’ personal information private while maintaining security.Health records are kept confidential while still being easily accessible when needed for treatment. Implementing a blockchain-based healthcare system is fraught with its own set of challenges. There are a number of challenges that must be overcome before blockchain technology can be widely adopted, including those related to the platform’s scalability and interoperability, ensuring compliance with norms and standards, and easing privacy fears. We looked at the potential of a healthcare system built on blockchain technology and concluded that more study is needed. The goals of these initiatives should be to improve healthcare providers’ and hospitals’ data security, privacy, and efficiency by addressing the challenges mentioned above.

## 9. Conclusions

In conclusion, tremendous progress has been made in the field of cybersecurity and privacy in cyber-physical systems (CPSs) since the introduction of the breakthrough privacy design utilizing consortium blockchain. Traditional encryption-based models, access control-based models, and centralized database-based models all have flaws and research gaps that need to be filled, and this novel solution fills those gaps and tackles the security challenges. Several important conclusions have surfaced as a result of doing a comparative examination of these benchmark models. While the conventional encryption-based methodology does protect the privacy of users’ data, it is not transparent and cannot be scaled to accommodate large-scale systems. Granularity, policy management, and context-aware access control are all areas that present difficulties for the paradigm that is based on access control. The centralized database-based architecture is susceptible to having a single point of failure, has limited scalability, and is dependent on having faith in a centralized authority. The proposed ground-breaking privacy design with consortium blockchain is able to address these difficulties because it offers storage that is both secure and decentralized, access control that is fine-grained, and auditability that is visible. It does so by capitalizing on the one-of-a-kind characteristics of blockchain technology, such as immutability, consensus processes, and smart contracts, in order to safeguard the confidentiality of CPS data and maintain its integrity. Overall, the breakthrough privacy design with consortium blockchain presents a viable approach for shielding secrets in CPSs, giving increased security, privacy, transparency, and accountability. It is a design that was made possible by the consortium blockchain. It does this by resolving the research gaps and security vulnerabilities that are present in the benchmark models, which prepares the way for a CPS environment that is both more safe and more respectful of users’ privacy.

## Figures and Tables

**Figure 1 sensors-23-07162-f001:**
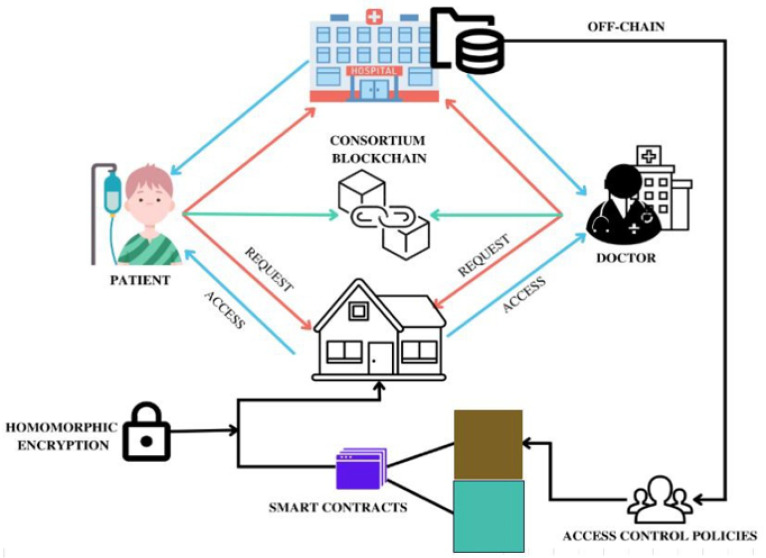
Proposed framework.

**Figure 2 sensors-23-07162-f002:**
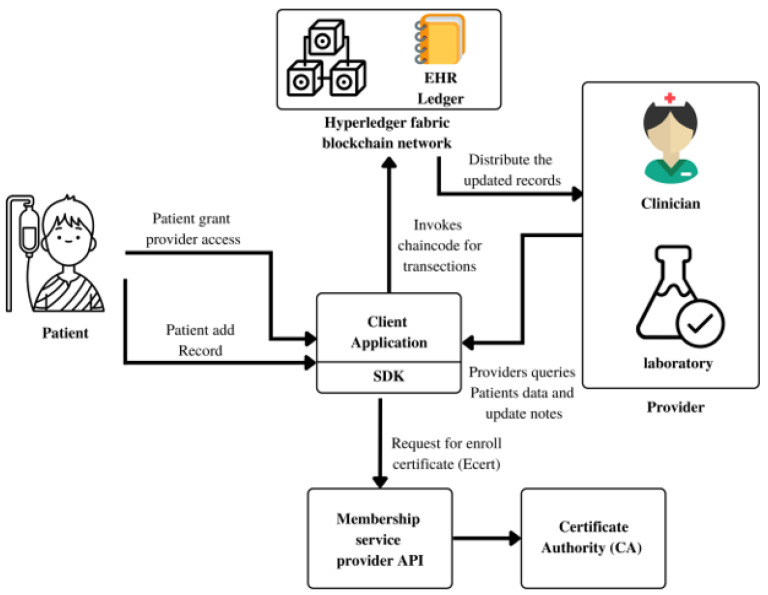
Proposed system architecture.

**Figure 3 sensors-23-07162-f003:**
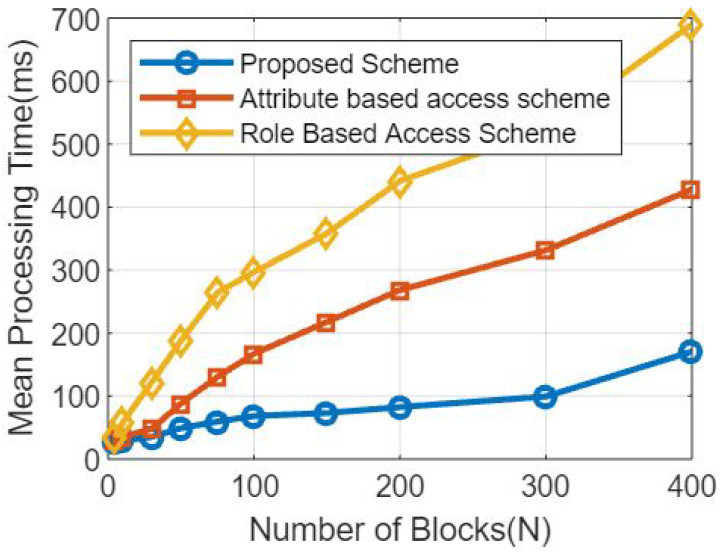
Analysis based on the amount of mean processing time and blocks used in a simulation.

**Figure 4 sensors-23-07162-f004:**
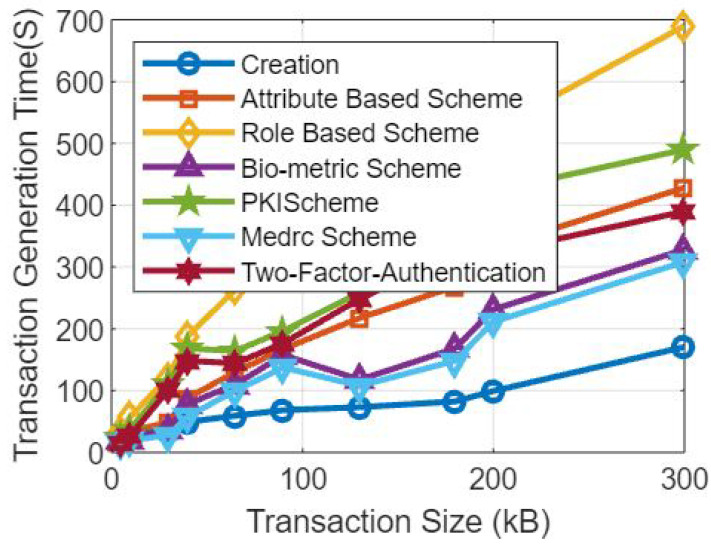
Simulation results based on the number of transactions and transaction generation time in seconds.

**Figure 5 sensors-23-07162-f005:**
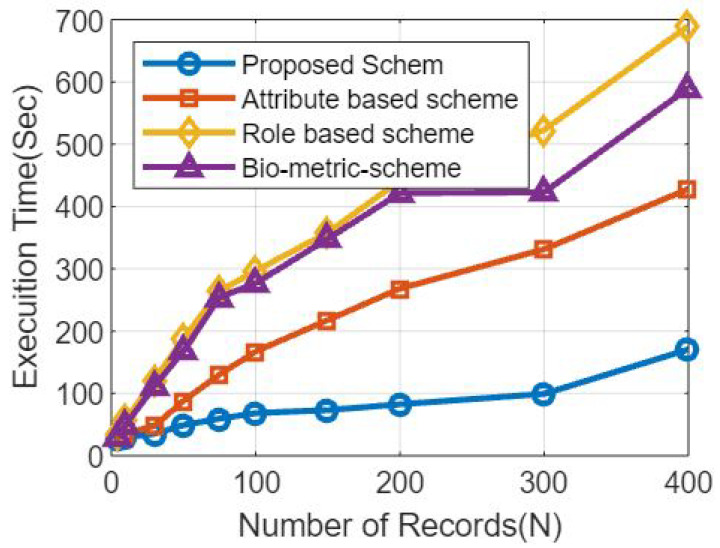
Analysis based on the number of records and the transaction times.

**Figure 6 sensors-23-07162-f006:**
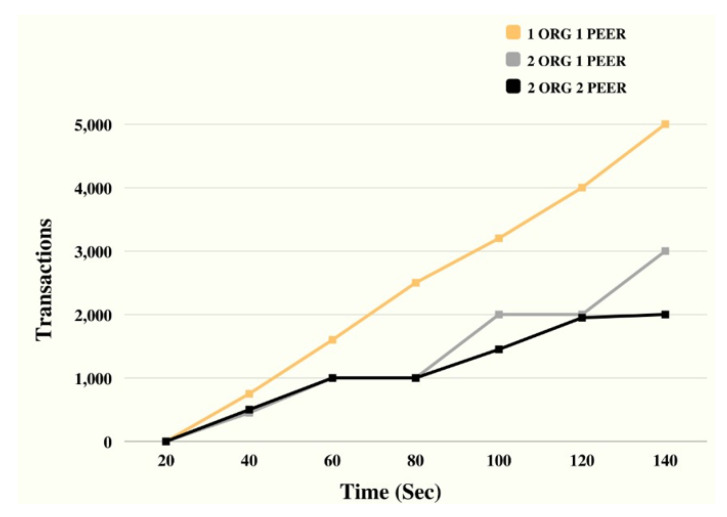
Simulation results based on the execution times in seconds and the number of transactions.

**Figure 7 sensors-23-07162-f007:**
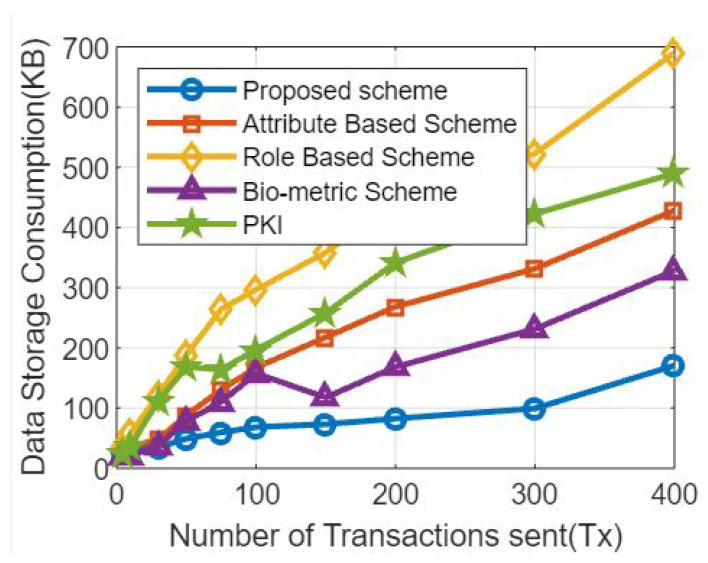
Comparative analysis based on the number of transactions and the storage consmuption in comparison with the benchmark model.

**Figure 8 sensors-23-07162-f008:**
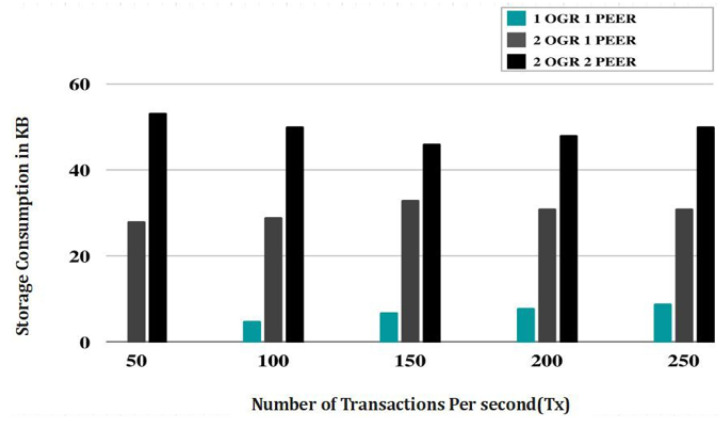
Simulation results based on the number of transactions and memory consumption.

**Figure 9 sensors-23-07162-f009:**
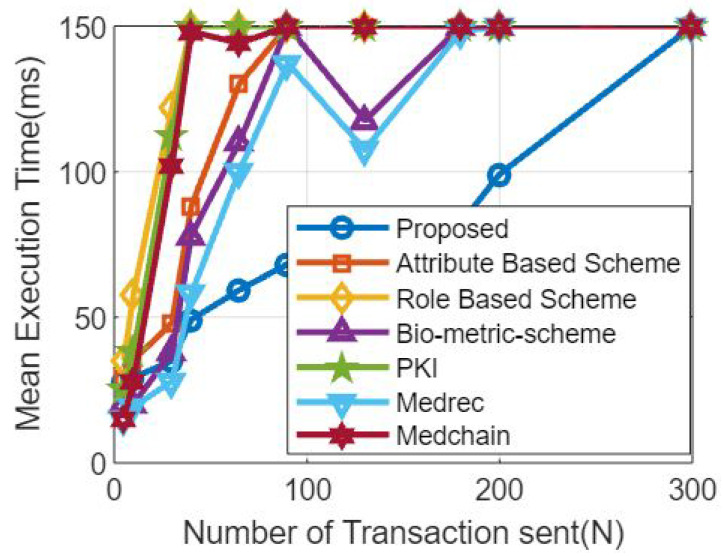
Simulation results based on the number of transactions and the Mean execuition Time.

**Table 1 sensors-23-07162-t001:** List of abbreviations.

Abbreviation	Description
CPSs	Cyber-Physical Systems
ACL	Access Control List
DB	Database
IoT	Internet of Things
PoW	Proof of Work
PoS	Proof of Stake
EHR	Electronic Health Record
GDPR	General Data Protection Regulation
HIPAA	Health Insurance Portability and Accountability Act

**Table 2 sensors-23-07162-t002:** Comparative analysis of benchmark models.

Model	Security Issues	Research Gaps	Problem	Solution
Traditional encryption-based model	Lack of data transparency	Limited scalability for large-scale systems	Difficulty in managing encryption keys	Explore the use of homomorphic encryption to allow data processing on encrypted data
	Vulnerability to key-based attacks	Inadequate protection against insider threats	Lack of adaptability to dynamic environments	Develop techniques for secure key management and continuous monitoring of user activities
Access control-based model	Limited granularity in access control policies	Complex management of access control rules	Insufficient support for context-aware access control	Investigate attribute-based access control (ABAC) with dynamic policy enforcement
	Difficulty in handling user revocation	Inability to address data sharing across multiple domains	Lack of fine-grained auditing capabilities	Explore the use of blockchain-based access control mechanisms and distributed ledgers
Centralized database-based model	Single point of failure	Potential data breaches due to centralization	Limited transparency and accountability	Investigate the use of distributed databases or decentralized storage systems
	Scalability limitations for large datasets	Dependency on trust in the centralized authority	Difficulty in ensuring data integrity	Explore distributed consensus algorithms for decentralized data management

**Table 3 sensors-23-07162-t003:** Simulation results—execution times and number of transactions.

Benchmark Model	Number of Transactions	Execution Time (Seconds)
Proposed	1000	15.6
[1]	1000	22.3
[5]	1000	18.9
[6]	5000	78.2
[13]	5000	105.9
[25]	5000	92.7
[25]	10,000	156.8
[8]	10,000	215.4
[38]	10,000	189.6

**Table 4 sensors-23-07162-t004:** Comparative analysis of security attacks.

Attack Type	Benchmark Model [12]	Benchmark Model [13]
Denial of service (DoS)	High vulnerability to DoS attacks due to insufficient network resources allocation	Effective DoS attack prevention mechanisms in place
Man-in-the-middle (MitM)	Vulnerable to MitM attacks due to weak encryption protocols	Robust encryption and authentication protocols to mitigate MitM attacks
Phishing	Lack of effective phishing detection and prevention mechanisms	Advanced phishing detection techniques implemented
Malware	Prone to malware infections and lacking effective malware detection	Robust malware detection and prevention mechanisms

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
